# Low-Grade Gliomas in Patients with Noonan Syndrome: Case-Based Review of the Literature

**DOI:** 10.3390/diagnostics10080582

**Published:** 2020-08-12

**Authors:** Mariachiara Lodi, Luigi Boccuto, Andrea Carai, Antonella Cacchione, Evelina Miele, Giovanna Stefania Colafati, Francesca Diomedi Camassei, Luca De Palma, Alessandro De Benedictis, Elisabetta Ferretti, Giuseppina Catanzaro, Agnese Pò, Alessandro De Luca, Martina Rinelli, Francesca Romana Lepri, Emanuele Agolini, Marco Tartaglia, Franco Locatelli, Angela Mastronuzzi

**Affiliations:** 1Department of Paediatric Haematology/Oncology, Cell and Gene Therapy, Bambino Gesù Children’s Hospital, IRCCS, 00165 Rome, Italy; mariachiara.lodi@opbg.net (M.L.); antonella.cacchione@opbg.net (A.C.); evelina.miele@opbg.net (E.M.); franco.locatelli@opbg.net (F.L.); 2School of Nursing, College of Behavioral, Social and Health Sciences, Clemson University, Clemson, SC 29634, USA; lboccut@clemson.edu; 3JC Self Research Institute of the Greenwood Genetic Center, Greenwood, SC 29646, USA; 4Neurosurgery Unit, Department of Neuroscience and Neurorehabilitation, Bambino Gesù Children’s Hospital, IRCCS, 00165 Rome, Italy; andrea.carai@opbg.net (A.C.); alessandro.debeneditcis@opbg.net (A.D.B.); 5Neuroradiology Unit, Department of Imaging, Bambino Gesù Children’s Hospital, IRCCS, 00165 Rome, Italy; gstefania.colafati@opbg.net; 6Department of Laboratories, Pathology Unit, Bambino Gesù Children’s Hospital, IRCCS, 00165 Rome, Italy; francesca.diomedi@opbg.net; 7Neurology Unit, Department of Neuroscience, Bambino Gesù Children’s Hospital, IRCCS, 00165 Rome, Italy; luca.depalma@opbg.net (L.D.P.); elisabetta.ferretti@uniroma1.it (E.F.); 8Department of Experimental Medicine, Sapienza University, 00161 Rome, Italy; giuseppina.catanzaro@uniroma1.it; 9Department of Molecular Medicine, Sapienza University, 00161 Rome, Italy; agnese.po@uniroma1.it; 10Medical Genetics Division, Fondazione IRCCS-Casa Sollievo della Sofferenza, 71043 San Giovanni Rotondo, Italy; a.deluca@css-mendel.it; 11Laboratory of Medical Genetics, Department of Laboratories, Bambino Gesù Children’s Hospital, IRCCS, 00165 Rome, Italy; martina.rinelli@opbg.net (M.R.); francescaromana.lepri@opbg.net (F.R.L.); emanuele.agolini@opbg.net (E.A.); 12Genetics and Rare Diseases Research Division, Bambino Gesù Children’s Hospital, IRCCS, 00165 Rome, Italy; marco.tartaglia@opbg.net; 13Department of Pediatric Hematology and Oncology Cell and Gene Therapy, Bambino Gesù Hospital, IRCCS, University La Sapienza, 00165 Rome, Italy

**Keywords:** pediatric brain tumor, cancer predisposition, Noonan syndrome, mTOR signaling, everolimus

## Abstract

Noonan syndrome (NS) is a congenital autosomic dominant condition characterized by a variable spectrum from a clinical and genetical point of view. Germline mutations in more than ten genes involved in RAS–MAPK signal pathway have been demonstrated to cause the disease. An higher risk for leukemia and solid malignancies, including brain tumors, is related to NS. A review of the published literature concerning low grade gliomas (LGGs) in NS is presented. We described also a 13-year-old girl with NS associated with a recurrent mutation in *PTPN11*, who developed three different types of brain tumors, i.e., an optic pathway glioma, a glioneuronal neoplasm of the left temporal lobe and a cerebellar pilocytic astrocytoma. Molecular characterization of the glioneuronal tumor allowed to detect high levels of phosphorylated MTOR (pMTOR); therefore, a therapeutic approach based on an mTOR inhibitor (everolimus) was elected. The treatment was well tolerated and proved to be effective, leading to a stabilization of the tumor, which was surgical removed. The positive outcome of the present case suggests considering this approach for patients with RASopathies and brain tumors with hyperactivated MTOR signaling.

## 1. Introduction

Noonan syndrome (NS, OMIM PS163950) is a genetically heterogeneous condition characterized by wide clinical variability. It is considered one of the prevalent non-chromosomal disorders affecting development and growth [1].

NS is characterized by reduced postnatal growth, distinctive facial traits (hypertelorism, downslanting palpebral fissures, ocular ptosis, high forehead, triangular face, low-set and posteriorly rotated ears and short and/or webbed neck), a variable spectrum of congenital heart defects (CHDs) and hypertrophic cardiomyopathy, learning difficulties, renal anomalies, lymphatic malformations, bleeding disorders and skeletal malformations [1,2,3]. The disorder overlaps clinically with Costello syndrome (CS), cardio–facio–cutaneous syndrome (CFCS), Noonan syndrome with multiple lentigines (NSML, known also as LEOPARD syndrome), Mazzanti syndrome, neurofibromatosis Type 1 (NF1) and Legius syndrome [2]. These disorders are related to germline mutations in the genes encoding proteins within the RAS-mitogen-activated protein kinase (MAPK) signaling pathway and have therefore been referred to as RASopathies [2,4]. NS is heterogeneous disorder, with causative germline mutations been reported in the “*PTPN11*, *SOS1*, *KRAS*, *NRAS*, *RAF1*, *BRAF*, *MEK1, RIT1, SOS2, LZTR1, MRAS* and *RRAS2* genes”. RASopathies are considered as cancer-prone disorders [5]. In NS, childhood juvenile myelomonocytic leukemia and other hematologic malignancies most frequently occur, followed by neuroblastoma, primary brain tumors and rhabdomyosarcoma. In the present work, we report the clinical management of a pediatric patient with *PTPN11*-related NS carrying three different low grade gliomas (LGGs) and reviewed the published cases in literature about pediatric LGGs in NS.

## 2. Case Presentation

The patient was born to non-consanguineous parents at 33 weeks after an non pathologic gestation. No abnormality was noted at birth; however, since the first month of life she showed slight development delay, reduced growth and dysmorphic facial features. Due to a holosystolic heart murmur detected in the second week of life, an echocardiography was performed revealing pulmonary valve stenosis. At one month, based on these clinical features, the patient was suspected to have NS and a genetic analysis performed using a dedicated gene panel (*BRAF*, *CBL*, *CDC42*, *HRAS*, *KRAS*, *MAP2K1*, *MAP2K2*, *NF1*, *NRAS*, *PTPN11*, *RAF1*, *RIT1, RRAS*, *SHOC2*, *SOS1*, *SOS2* and *SPRED1*) revealed the pathogenic *PTPN11* variant NM_002834.3 (c.922A>G, p.Asn308Asp), one of the most common mutations associated with NS [3]. Sanger sequencing detected the same variant in her mother, confirming cosegregation of the mutation with disease in this family (Figure 1A,B). In addition, a maternally transmitted missense variant in *SPRED1* (c.182G>A, p.Arg61His; NM_152594.3) was identified. This missense change was classified as “likely benign”, based on the American College of Medical Genetics criteria (PM1, BS2, BP1, BP4). Family history was positive for pulmonary stenosis in one brother, who died at the age of 13 years for acute lymphoblastic leukemia. No clinical report or biologic material to be analyzed was available.

The patient′s family has provided informed consent to the sharing of clinical information and images for research purposes. Such consent has been designed in accordance with the internal policy approved by the ethical committee of the Bambino Gesù Hospital (approval code 347/RA, n°1353/2017).

The proband was hospitalized at 18 months of age because of a prolonged febrile seizure. At admission, brain magnetic resonance imaging (MRI) revealed few small areas of hyper-intense signal on T2-weighted images without contrast enhancement after gadolinium injection involving the frontal and temporal right lobes and the splenium of corpus callosum. The patient underwent periodic follow-up, during which a second MRI analysis showed a new suprasellar–chiasmatic lesion suspected of optic pathway glioma; subsequently, a further small lesion with cystic component and a solid enhancing part on the right cerebellar hemisphere was also reported (Figure 2A).

When the patient was 6 years old, she presented with a cluster of focal seizures characterized by sudden abdominal pain, vomiting, oro-alimentary automatism with subsequent stiffening of the left hemisoma and then sleepiness in the post-ictal period. She was therefore referred to our center where video-electroencephalography documented right temporo-mesial seizures. MRI confirmed the presence of multiple lesions, hyperintense in T2 derived images, mainly affecting the optic diencephalic region, the right temporo–mesial structure, the motor cingulate and right cerebellum (Figure 2A). Treatment with valproic acid and lacosamide did not allow seizure control. A new MRI showed progression of the cerebellar lesion (Figure 2A).

The appearance of multiple *café-au-lait* spots, increased in number and size with age, multiple lentigines and the presence of an optical–diencephalic lesion detected by MRI, led to suspect the possibility of concomitant NF1. Molecular data were reevaluated and an MLPA test for detection of intragenic *NF1* structural rearrangements was performed with negative result. A clinical exome sequencing of the proband and her parents was also performed using DNA obtained from peripheral blood to exclude additional germline variants in genes associated with cancer predisposition syndromes. Library preparation and exome capture of patient with her parents were performed by using a custom gene panel (Twist Bioscience). The “BaseSpace pipeline” (Illumina, https://basespace.illumina.com/) and the “TGex software” (LifeMap Sciences, Inc., Alameda, CA 94501, USA) were used for the variant calling and annotating variants, respectively. Data analysis did not reveal any clinically/functionally variant related to cancer-prone conditions apart from the previously identified sequence changes in *PTPN11* and *SPRED1*.

Due to the persistence of seizures, at the age of 9 years, the patient underwent right craniotomy and standard anterior temporal resection, including mesial structures (uncus, amygdala and hippocampus). At the end of the resection, the lateral chiasmatic region was exposed and a limited biopsy of the optic pathway lesion was performed. No complication occurred in the postoperative period; currently (5 years after surgery), the patient is seizure-free, without requirement of anti-epileptic therapy. Histology of the temporal lesion was diagnostic for a glioneuronal neoplasm (WHO I). Immunohistochemistry (ICH) was negative for the *BRAF* V600E point mutation and positive for phosphorylated MTOR (Ser2448), indicating the activation of the MTOR pathway (Figure 3A).

After few months, due to the increased volume of the cerebellar lesion, with initial hydrocephalus, occipital craniotomy was performed with excision of the cerebellar cystic neoplasm that was histologically characterized as a cerebellar pilocytic astrocytoma (WHO I) (Figure 3B). At 11 years, brain MRI showed a volume increase of the chiasmatic lesion adjacent to the temporal resection margin, associated with visual impairment. Therefore, treatment with the MTOR inhibitor, everolimus, was started at the dose of 5 mg/day [4], subsequently reduced to 2.5 mg/day due to signs of toxicity (mouth ulcers). Follow-up MRI were performed every six months; the last one, carried out when she was 13-year-old, showed stable disease at 27 months since the beginning of therapy (Figure 2B). Frequent blood test evaluations were performed to exclude the development of leukemia.

## 3. Discussion

Similar to other RASopathies, certain mutations causing NS may predispose to cancer [5]. The risk of cancer in NS is evaluated at 4% by the age of 20 [6,7,8]. The underlying molecular mechanism involves the upregulation of the RAS–MAPK signaling pathway; such dysregulation leads to increased cellular proliferation which can lead to tumor development in those cells where additional driver somatic mutations occur. Somatic mutations in the *PTPN11* gene are found in 35% of sporadic juvenile myelomonocytic leukemia (JMML) and in other hematological malignancies [9,10]. Similarly, somatic *PTPN11* mutations have been detected with variable prevalence in some solid tumors, such as lung, liver and colorectal cancer, thyroid carcinoma, bladder carcinoma, neuroblastoma, rhabdomyosarcoma, melanoma and brain tumors [11,12].

NS can be caused by germline mutations in several genes related to the RAS–MAPK pathway cascade (i.e., *PTPN11*, *SOS1*, *SOS2*, *RAF1*, *KRAS*, *NRAS*, *RIT1*, *MRAS*, *RRAS2* and *LZTR1)*. A gain-of-function mutations in *PTPN11*, which encodes the SHP2 protein, causes 50% of cases of NS. SHP2 is a non-receptor protein tyrosine phosphatase (PTP), positively controlling RAS function. The protein has a complex regulatory mechanism controlling its subcellular localization and activation. Both germline and somatic *PTPN11* mutations are activating and enhance SHP2 function by destabilizing the catalytically inactive conformation of the phosphatase, resulting in increased signal flow through the RAS–MAPK pathway [13].

Patients with NS have a significant risk to develop JMML, embryonal rhabdomyosarcoma and brain tumors [7]. In particular, central nervous system (CNS) glial tumors are known to be a result of RASopathies [5,7]. A recent study conducted in France showed the correlation between JMML and NS in a cohort of 641 patients with germline *PTPN11* mutations. Somatic mutations in those genes are related to up to 20% of all sporadic hematologic and solid malignancies [14]. Germline mutations in those genes underlying tumor formation seams to determine enhanced signaling activity, although in a less consistent way than seen in corresponding somatic mutations. It is hypothesized that those activating somatic gene mutations causing cancer are not seen in the germline because they may cause embryonal lethality [15].

We reviewed the literature for cases of pediatric LGGs reported in patients with NS. PubMed and Google scholar were used for publications. Investigated terms included “Noonan” OR “PTPN11” AND “dysembryoplastic,” “pilocytic,” “medulloblastoma,” “oligodendroglioma,” “glioneuronal,” “astrocytoma,” “glioma,” “ependymoma,” “craniopharyngioma,” “papilloma,” or “tumor”. We included reviews, case reports, case series and case report abstracts.

We analyzed 30 cases of patients with NS and brain tumors carrying *PTPN11* mutation (Table 1). Most of them were glial or glioneuronal tumors. In a recent review the authors identified 22 cases of *PTPN11*-driven NS patients with brain tumors: mainly “dysembryoplasic neuroepithelial tumors” (DNETs), but also “medulloblastomas”, “oligodendrogliomas”, “astrocytomas”, “pilocytic astrocytomas (PA)”, “gliomas” and “mixed glioneuronal tumors” [16]. Siegfried at al. updated with 25 identified brain tumors [17]. Two other patients were subsequently reported [18], the first showing a dysembryoplastic neuroepithelial tumor, the second with a pilocytic astrocytoma. Finally, E-Ayadi et al. described two cases of anaplastic astrocytoma [19]. Out of the twenty-eight published cases, twenty-four were LGGs and glioneuronal tumors, most of which were DNETs. Of note, while PA are estimated the most common pediatric brain tumors and glioneuronal tumors are 30% of pediatric brain neoplasms, DTNs are rare (<1%), but account for approximately 40% of the brain tumors reported in NS [20]. Despite these findings, El-Ayadi et al. have proved that HGG may occur in NS [19]. Although our patient had low-grade gliomas, the peculiarity of our case is that she developed three different types of brain tumors at the same time, without any hematological disease. Another particular feature of our patient is that she had a positive genetic test for NS and, at the same time, she developed phenotypic characteristics also present in NF1, such as *café-au-lait* spots see [21], lentigines, in association with an optical-diencephalic lesion, although phenotypic overlapping is not uncommon for patients affected with RASopathies [22,23]. Nevertheless, the case was negative on clinical exome sequencing including NF1, but though she could have an undetected mutation that does not affect the coding part of the gene. To the best of our knowledge, no similar cases have been reported in the literature. The evidence provided by this and previous similar cases suggests the need for a close follow-up of patients affected by RASopathies starting from an accurate medical history and clinical examination aimed at identifying any characteristics referring to the RASopathy. Patients with a cancer predisposition syndrome should be evaluated periodically for specific types of tumors known to be frequent in this population (such as leukemia, lymphoma, neuroblastoma, rhabdomyosarcoma, brain tumors and bladder carcinoma) based on clinical symptoms, regular physical examinations and complete blood counts and, considering the high brain tumor risk, a regular screening protocol with brain MRIs may be implemented [24]. The early detection of cancer should positively influence patient outcome, just as a long follow-up is essential for those patients who have already developed a tumor. Characterizing the underlying tumor molecular alterations can facilitate the development of targeted therapies to improve efficacy of the treatment, quality of life and long-term survival in those patients.

Syndromic patients represent a particular challenge to the neurosurgeon. When defining the surgical flap to approach to a specific lesion the possibility of further surgeries for new tumors must be considered setting a global strategy to possibly access the whole brain limiting the need for inappropriate subsequent skin incisions. Our patient presented a temporal lobe lesion and a separate optic pathway tumor on the same side. Biopsy of optic pathway lesions is not generally encouraged unless imaging data suggest atypical features or there is need to explore molecular targets [37]. Morbidity of surgical debulking or biopsy of optic pathway lesions has been shown to be acceptable with modern surgical techniques [38].

Considering the overall growth of all intracranial tumors in our patient, the high value of a targeted therapy approach in a syndromic child with higher chance to suffer from chemotherapy toxicity [39], reference and the chiasmatic lesion needed treatment, we decided to perform a limited biopsy to confirm histology and possible molecular targets. Performing the biopsy at the end of the temporal resection not only avoided a second craniotomy later on, but also allowed us to take advantage of the temporal mesial resection that exposed the area of interest avoiding additional dissection. This case demonstrates the benefit of a dedicated multidisciplinary team including neurosurgeons, neuro-oncologists, neuroradiologists and pathologists in the optimal management of such complex cases.

Another important aspect of our case is the positive response to an mTOR inhibitor. While the functional link between the identified *PTPN11* mutation and MTOR hyperactivation needs further study, the present finding suggests a possible molecular target for a new therapeutic strategy and that assessment of MTOR activation should be evaluated in brain tumors of patients with NS or a related RASopathy. Our findings are in line with previous reports suggesting that the PI3K/mTOR signaling may mediate the oncogenic signal elicited by enhanced SHP2 function. These findings indicate a therapeutic potential of rapamycin analogs for *PTPN11* mutation-associated cancers [39,40].

## 4. Conclusions

The success of the rapalog everolimus in treating subependymal giant cell astrocytomas (SEGAs) associated with tuberous sclerosis proved that this drug could shrink mTOR-driven tumors [15,41]. Everolimus may have activity in other pLGG. However, it is advisable to confirm the improper activation of the AKT–mTOR signaling pathway by molecular test before administering an mTOR inhibitor, because children with NS respond to chemotherapy treatment, but they are more likely to experience severe toxicity [42]. In our case, everolimus was well tolerated. For this reason, the use of everolimus and other targeted therapies in the treatment of genetic cancer syndromes provides significant opportunities for improved quality of life for patients with RASopathies. However, further research is needed to discern appropriate use of available pharmacologic agents as well as to develop more effective targets with reliable, durable responses for each of these conditions. We acknowledge that supplemental considerations may stem from further characterizations of LGGs in patients with NS and *PTPN11* variants, however, we feel that the frequency of these tumors in NS is still too low to allow solid conclusions. The positive response to everolimus observed in our patient may provide an ex adiuvantibus piece of evidence for a pathogenic link between the RAS–MAPK and PI3K/mTOR pathways, activated by *PTPN11* variants, but, as we stated in our Discussion, the functional link between the identified *PTPN11* mutation and MTOR hyperactivation needs further study.

## Figures and Tables

**Figure 1 diagnostics-10-00582-f001:**
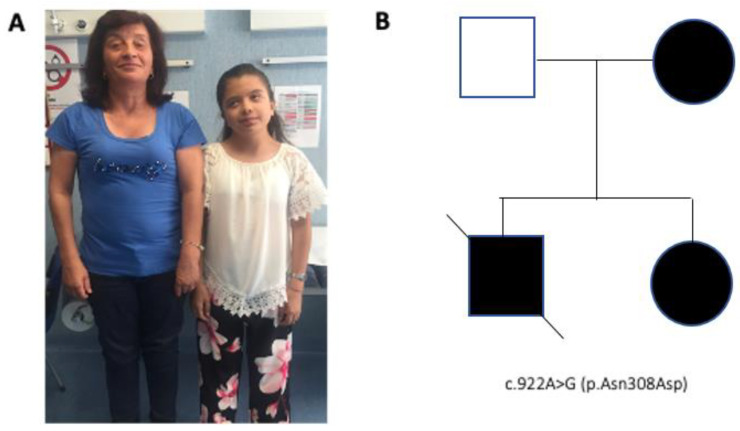
(**A**) Patient (right) at age 13 years and her mother (left) show typical features of NS, note the short stature and common facial features (ptosis, hypertelorism, downslanting palpebral fissures and low-set ears) and a short webbed neck; (**B**) pedigree of the family in which NS cosegregated with the recurrent *PTPN11* mutation.

**Figure 2 diagnostics-10-00582-f002:**
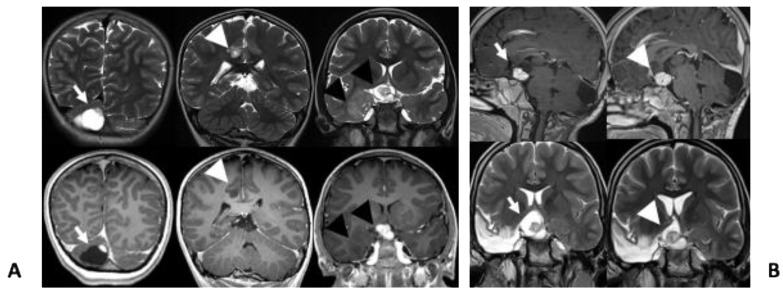
(**A**) Coronal T2w (top row) and T1w (bottom row) images show cystic lesion in the cerebellum with a peripheral enhancing nodule (arrows), a small hyperintense lesion without contrast-enhancement involving the right gyrus cinguli (arrow heads). There is also a suprasellar/chiasmatic lesion (black arrows) with avid contrast-enhancement and cystic components, consistent with optic pathway glioma and a hyperintense and non enhancing lesion involving the right mesial temporal lobe (black arrow heads); (**B**) sagittal GdT1w (top row) and coronal T2w (bottom row) images show optic pathway glioma before (arrows) and during treatment (arrow heads).

**Figure 3 diagnostics-10-00582-f003:**
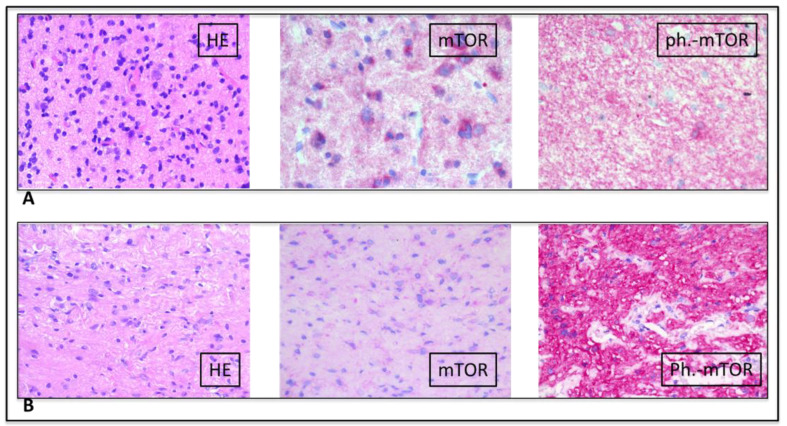
(**A**) (upper raw) (low-grade glioneural tumor): HE: Low intensity glial proliferation with intermingled few dysplastic ganglion cells. Mild nuclear atypia and low proliferation index. mTOR: mild cytoplasmic positivity of neoplastic cells. Phospho–mTOR: strong cytoplasmic positivity of neoplastic cells; (**B**) (lower raw) (pilocytic astrocytoma): HE: Pilocytic glial proliferation in a fibrillary background with evidence of some Rosenthal fibers. Mild nuclear atypia and low proliferation index. mTOR: mild cytoplasmic positivity of neoplastic cells. Phospho–mTOR: Strong cytoplasmic positivity of neoplastic cells.

**Table 1 diagnostics-10-00582-t001:** Primary brain tumors reported in Noonan syndrome.

No.	Reference	Age	Gender	Noonan Syndrome Diagnosis	Brain Tumor Diagnosis	Location
1	McWilliams et al. [16]	8	M	*PTPN11* p.Glu139Asp	DNET	Temporal lobe and cerebellum
2	Jongmans et al. [6]	10	Ukn	*PTPN11* c.179G > C; p.Gly60Ala	DNET	Temporal lobe
3	Selter et al. [25] Pellegrin et al. [17]	13	M	*PTPN11* exon 3 **	DNET	Left parietal lobe
4	Bendel et al. [26]	17	M	*PTPN11* c.174C > G; p.Asn58Lys	DNET	Occipital cortex
5	Bendel et al. [26]	37	M	Maternal uncle of Case 4	DNET	Unknown
6	Delisle et al. [27]	12	M	*PTPN11* mutation **	DNET	Temporal lobe and thalamus
7	Krishna et al. [18]	9	M	*PTPN11* p.Asp61Gly	DNET	Temporal lobe and cerebellum
8	Pellegrin et al. [17]	13	M	N/A	DNET	Right parieto-occipital cortex
9	Pellegrin et al. [17]	13	M	*PTPN11* exon 3 mutation **	DNET (MRI)	Left parietal lobe
10	Kratz et al. [8]	6	F	*PTPN11* p.Asn308Asp	DNET	Unknown
11	Siegfried et al. [17]	16	M	*PTPN11* c.922A > G; p.Asn308Asp	DNET	Left temporal and frontal lobe, right thalamus
12	Rankin et al. [26]	10	M	*PTPN11* c.1403C > T; p.Thr468Met	Medulloblastoma	Cerebellum
13	Jongmans el al [6]	18	F	*PTPN11* c.417G > C; p.Glu139Asp	Oligodendroglioma	Hypothalamus
14	Sherman et al. [28]	6	M	*PTPN11* c.172A > G; p.Asn58Asp	Low-grade mixed glioneuronal tumor	Suprasellar cisterns, sella turcica and hypothalamus and diffuse leptomeningeal disease
15	Schuettpelz et al. [29]	8	M	*PTPN11* c.1471C > T and c.1472C > T; p.Pro491Phe	Pilocytic astrocytoma	Sellar/suprasellar with extension to prepontine region to the level of the pontomedullary junction
16	Fryssira et al. [30]	11	F	Clinical diagnosis	Pilocytic astrocytoma	Sellar/suprasellar
17	De Jongo et al. [31]	21	M	*PTPN11* mutation **	Multiple indeterminate lesions on MRI	Multiple: supratentorial, infratentorial, cortical and subortical, thalamus
18	Karafin et al. [32]	18	M	Clinical diagnosis	Rosette forming glioneuronal tumor	Fourth ventricle
19, 20, 21	Rush et al. * [17]	Ukn	M	*PTPN11* mutation **	Low grade astrocytoma	Suprasellar and thalamic region
22	Kratz et al. [19]	7	M	*PTPN11* p.Gly60Ala	Pilocytic astrocytoma	Right optic nerve
23	Takagi at al. [33]	20	M	Clinical diagnosis	Glioma	Unknown
24	Standford et al. [34]	16	M	Clinical diagnosis	Pilocytic astrocytoma	Intramedullary spinal cord lesion involving the cervical medullary junction and descending to the C2-C3 disc space level
25	Nair et al. [35]	14	M	*PTPN11* c.417G > C in exon 4	Pilomyxoid astrocytoma	Right optic nerve
26	Bendel and Pond. [19]	14	F	*PTPN11* c.922A > G; p.Asn308Asp	High grade glioma	Left brainstem/cerebellar peduncle
27	Martinelli et al. [36]	24	Ukn	*PTPN11* c.64A > G; pThr22Ala	Oligodendroglioma grade II	Unknown
28	El Ayadi et al. [19]	14	F	*PTPN11* c.922A > G; p.Asn308Asp	Anaplastic astocytoma	Left brainstem/cerebellar peduncle
29	El Ayadi et al. [19]	9	M	*PTPN11* c.5C > T; p.Thr2lle	Anaplastic astocytoma	Third ventricle, cerebellum and fornix
30	Our case	9	F	*PTPN11* c.922A > G; p.Asn308Asp	Pilocytic astrocitoma and glioneuronal tumor	Cerebellum Right temporal lobe

* patients 19, 20 and 21 were from a case series abstract of two males and one female. Two patients were deaf and possibly had NSML. DNET—dysembryoplasic neuroepithelial tumor. ** Mutation unknown.

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
