# Peer review of "Low-Grade Gliomas in Patients with Noonan Syndrome: Case-Based Review of the Literature"

_diagnostics, 2020, doi:10.3390/diagnostics10080582_

Round 1
Reviewer 1 Report
Excellent and original paper.
The Paper reported an interesting case in using a novel treatment of everolimus in patient with Noonan syndrome PTPN11 mutated gene who was affected by 3 different brain tumors. Using this treatment, they demonstrated a stabilization of these lesions with an opportunity for a surgical approach Background, case report and discussion was adeguate. An esaustive review of recent literature has been also reported. English language is correct In my opinion, this paper can be accepted in this version without any concerning.
Author Response
Thank you very much for what the reviewer wrote.
Reviewer 2 Report
The authors describe a young patient with NS and three brain tumors (low grade gliomas) , the investigations that heve been performed and the therapies that were given.
It is an important study with an important conclusion on using everolimus in patients with a RASopathy with brain tumors.
My comments:
Abstract
Line 48: autosomal dominant condition ( not a disease)
Case
Line 99:in the brother the diagnosis NS was not considered? Has he died before the patient was born? The condition in the brother deserves more comments. What investigations were performed in this brother and with what results? Brain MRI done in this brother?
Line 112: patient was 6 years old….
Discussion
Line 217: café-au-lait spots and lentigines are very frequent dermatologic manifestions of NS, present in about 80% of cases, see Bessis et al, Br J Dermatol, 2019 Jun;180(6):1438-1448. I miss this reference in the manuscript.
The optical-diencephalic lesion is a characteristic more related to NF1 than NS, but the dermatologic features are not.
Line 221: The authors state here that there is need for close follow-up of patients affected by RASopathies with accurate medical history and clinical examination… Here also the family history should be mentioned. The family history of their patient needs some comments. The mother has obviously NS and the brother had it, but the authors give no comments on missing the diagnosis NS in these two family members before .
Line224: Here the authors state that “ patients with a cancer predisposition syndrome should be evaluated periodically for specific types of tumours known to be frequent in this population (such as leukaemia, lymphoma, neuroblastoma, rhabdomyosarcoma, brain tumours, and bladder carcinoma) based on clinical symptoms, physical examination, and laboratory findings.”
What sort of laboratory investigations do they suggest? And why? References??
Line 241: ….”.in a syndromic child with higher chance to suffer from chemotherapy”..... : I miss the reference(s) here.
Conclusion
In the conclusion the authors only discuss the therapy (everolimus) that was used in their patient. Are there no other conclusions to make on their study of LGGs in patients with NS and PTPN11 mutations?
Line 260: …because children with NS respond to chemotherapy treatment but they are more likely to experience severe toxicity.” Again no refence(s) for this statement.
Author Response
Thank you very much for the reviewer's helpful comments and timely suggestions. Here are the answers to some comments. The modified manuscript is attached.
1) Line 99: Unfortunately, the brother died before the patient was born. He was treated at another hospital from which we have not received any clinical report or biological material to be analyzed.
2) Line 217: Another particular feature of our patient is that she had a positive genetic test for NS and, at the same time, she developed phenotypic characteristics also present in NF1, such as café-au-lait spots see [33], lentigines, in association with an optical-diencephalic lesion, although phenotypic overlapping is not uncommon for patients affected with RASopathies [34,35].
3) Line 221: Unfortunately, no clinical records have come down to us about the brother. Her mother’s genetic analysis was performed at the same time of the patient and not previously. The mother also denied any cardiological abnormalities.
4) Line224: “regular physical examinations and complete blood counts and, considering the high brain tumor risk, a regular screening protocol with brain MRIs may be implemented”. Recommendations for Cancer Surveillance in Individuals with RASopathies and Other Rare Genetic Conditions with Increased Cancer Risk. Villani A, Greer MC, Kalish JM, Nakagawara A, Nathanson KL, Pajtler KW, Pfister SM, Walsh MF, Wasserman JD, Zelley K, Kratz CP.Clin Cancer Res. 2017 Jun 15;23(12):e83-e90.
5) Conclusion: We acknowledge that supplemental considerations may stem from further characterizations of LGGs in patients with NS and PTPN11 variants, however, we feel that the frequency of these tumors in NS is still too low to allow solid conclusions. The positive response to everolimus observed in our patient may provide an ex adiuvantibus piece of evidence for a pathogenic link between the RAS-MAPK and PI3K/mTOR pathways, activated by PTPN11 variants, but, as we stated in our Discussion, the functional link between the identified PTPN11 mutation and MTOR hyperactivation needs further study.
